# Films of Bacterial Cellulose Prepared from Solutions in *N*-Methylmorpholine-*N*-Oxide: Structure and Properties

**Igor S. Makarov** [1,*], **Gulbarshin K. Shambilova** [2], **Markel I. Vinogradov** [1], **Pavel V. Zatonskih** [1], **Tatyana I. Gromovykh** [3], **Sergey V. Lutsenko** [3], **Natalia A. Arkharova** [4] and **Valery G. Kulichikhin** [1]

1   A.V. Topchiev Institute of Petrochemical Synthesis, Russian Academy of Sciences, 119991 Moscow, Russia; m.i.vinogradov1989@gmail.com (M.I.V.); zatonskikh@ips.ac.ru (P.V.Z.); klch@ips.ac.ru (V.G.K.)
2   Department of Chemistry and Chemical Technology, Atyrau State University named after Kh. Dosmukhamedov, Atyrau 060011, Kazakhstan; shambilova_gulba@mail.ru
3   Department of Biotechnology, I.M. Sechenov First Moscow State Medical University, 119991 Moscow, Russia; gromovykhtatyana@mail.ru (T.I.G.); svlutsenko57@mail.ru (S.V.L.)
4   A.V. Shubnikov Institute of Crystallography, Federal Research Center Crystallography and Photonics, Russian Academy of Sciences, 119333 Moscow, Russia; natalya.arkharova@yandex.ru
*   Correspondence: makarov@ips.ac.ru

**Abstract:** In the present study, one of the possible methods of the bacterial cellulose processing is proposed via its dissolution in *N*-methylmorpholine-*N*-oxide using the stage of mechano-chemical activation of the solid polymer–solvent system. Preliminary solid-phase activation is apparently a decisive factor affecting the dissolution rate of bacterial cellulose in *N*-methylmorpholine-*N*-oxide. The effects of bacterial cellulose concentration, solvent nature, degree of polymerization and temperature on dissolution time were studied. The rheological behavior of the solutions does not change at 120 °C for at least half an hour that allowed us to process such solutions for films preparation. The films from these solutions by means of dry-wet jet spinning in aqueous coagulant were formed. The structure of the nascent cellulose and formed films was tested by the X-ray diffraction method and SEM. The thermal behavior of the films revealed an increase in the carbon yield for the formed films compared to the nascent bacterial cellulose. The process of film pyrolysis is accompanied by exothermic effects, which are not typical for wood cellulose. Some reasons of such thermal behavior are considered.

**Keywords:** bacterial cellulose; *N*-methylmorpholine-*N*-oxide; mechano-chemical activation; cellulose solution; rheology; film formation; structure

## 1. Introduction

The process of producing cellulose by bacteria discovered by Brown over a hundred years ago [1] remained without significant interest for many years. Only after half a century the bacterial cellulose (BC) started to attract attention of the scientific and industrial groups due to its unique properties, which are hydrophilicity, biocompatibility with a living system, high values of mechanical characteristics, etc. [2]. Currently, BC has a very wide range of applications, including additives to chocolate in the food industry, bone tissue engineering, antimicrobial wound dressing in medicine and even sensitive diaphragms for microphones in electronics [3,4]. In addition, unlike the wood pulp, the BC does not contain low molecular weight impurities such as lignin and hemicellulose [5]. As a result, a content of alpha-cellulose in BC reaches 97% [6,7]. The other features of BC are high polymerization degree, which

exceeds significantly the corresponding values for wood cellulose, and high content of crystalline phase [8–10].

From the list of cellulose producer bacteria: *Acetobacter*, *Azotobacter*, *Rhizobium*, *Pseudomonas* and others, the most effective producers of BC are *A. xylinum*, *A. hansenii* and *A. pasteurianus* [11]. Synthesis of BC begins with the generation of glucan chains from nutrient medium consisting of aqueous solution of sugar, fructose (maltose, xylose, starch and others) and oxygen within the producer and its extrusion out through tiny pores (holes) of the external surface of bacteria. The BC production process can be compared with natural silk formation or even wet spinning of fibers, where the dope leaves the spinneret into the coagulation bath [12]. The supramolecular structure of BC is a system of strips consisting of microfibrils. The average width of the strip is 80 nm, the thickness varies from 3 to 4 nm, and the length can reach up to 10 μm [13,14]. BC strips form a dense net, providing large values of the specific inner surface of the gel-like films (not less than 500 m$^2$/g) retaining large amounts of water [15].

As in the case of wood cellulose, the choice of direct solvents for BC is a non-trivial task. The reason is the system of inter- and intramolecular hydrogen bonds, which are responsible for the package density and structure of the polymer [16]. Among the described direct solvents for BC can be found ionic liquids [17,18], LiCl/DMAc [19], *N*-methylmorpholine-*N*-oxide (NMMO) [10,20,21], aqueous solutions of zinc chloride [22], etc.

The concentration of dissolved BC in the above solvents varies from 0.1% to 7%. An increase in the polymer content has limitations because of very high viscosity, which is caused by an extremely high degree of polymerization (DP) of the BC. The use of cellulose with a DP of more than 1500 allows obtaining solutions only with a few percent of polymer content. In addition, it is necessary to note the time of preparation of BC solutions. The duration of dissolution, depending on the dissolving system, can reach several (up to 3) days [17–22]. Long time of preparing solutions in some solvents is accompanied by necessity to use high temperatures. Preparation of solutions of BC in NMMO monohydrate requires heating of the system up to 90 °C and holding at this temperature during 12 h [20]. It has been reported [23,24] that NMMO and cellulose can decompose with an increase of temperature and time of preparation of solutions. Therefore, it is necessary to choose the right temperature-time conditions for preparing solution of BC in NMMO in order to exclude the destruction of components. Destruction of the solvent and the polymer affects not only the quality of the solution and consequently fibers or films, but also complicates the further regeneration of the solvent.

Based on high donor activity of N-O bonds in NMMO even in the solid state, the method of the solid-phase activation of the solvent–polymer system was developed by our group [25]. After such treatment the solid solution forms, which transforms to a viscoelastic state by simple heating to the melting point of the solvent. The concentrated solutions of the wood cellulose were obtained by this method [26], and series of high-tenacity fibers and films were spun from them [27].

The BC gives a unique opportunity to perform similar work on virtually defectless, pure, uniform in sense of molecular weight distribution and high molecular weight object. The objective of this paper is to develop a method for producing cellulose films from BC solutions in NMMO, which would render it possible to obtain "concentrated" polymer solutions with high DP values for relatively short periods of time. Special attention will be devoted to the analysis of rheology and morphology of concentrated solutions of BC, as well as features of the film formation and investigation of their structure and morphology. Successful carrying out of this research will help us to compare the basic characteristics of the nascent BC and processed films, and also get a new class of materials with unique properties inherent in nascent BC.

## 2. Materials and Methods

### 2.1. Materials

The *Gluconacetobacter hansenii* GH-1/2008 strain, not toxic and not zoopathogenic for higher homeothermic animals and humans, was used to obtain the BC [28]. Bacterial cellulose was obtained by stationary cultivation of *Gluconacetobacter hansenii* GH-1/2008 (VKPM V-10547) in modified Hestrin–Schramm medium (H5) previously described in [29], with the following composition (g/L): glucose—20.0, peptone—5.0, yeast extract—5.0, $Na_2HPO_4$—2.7, $KH_2PO_4$—2.0, citric acid monohydrate—1.15. Glucose (pure, pharma grade), $Na_2HPO_4$ and $KH_2PO_4$ were supplied by Sigma-Aldrich (Saint Louis, MO, USA). Powdered meat peptone with nitrogen content of more than 3.0% and yeast extract were purchased from Dia-M (Russia). In order to obtain seeding material, *G. hansenii* was cultivated in the described medium at 28 °C using the rotational shaker IKA KS 4000 ic for 24 h. BC films were obtained by means of cultivation of the producer in the stationary regime in the medium for 15 days at 28 ± 2 °C in 25 cm × 50 cm cuvettes it synthesizes 13.0 ± 0.5 g/L of the dry BC.

It should be noted that the producer of bacterial cellulose *G. hansenii* belongs to the group of Gram-negative bacteria that form endotoxins in the cell wall [30]. Therefore, to completely remove cells and endotoxins, the producer of the film was washed for 3 days with a 1 M NaOH solution at 25 °C, replacing the solution every 24 h. Then the film was washed repeatedly with distilled water to a pH of 7.0. *Limulus amoebocyte* reagent was used to control the endotoxins of the cell wall of *G. hansenii*. The bacterial endotoxin content was determined with a gel-clot test using *Limulus amoebocyte* (*Limulus polyphemus*) and lysate (LAL reagent, Limus Amebocyte Lysate, Reagent, US License No. 1197, Charles River Endosafe, USA) [31,32].

The degree of polymerization of BC was determined by measurement of the intrinsic viscosity of solutions in cadoxen according to GOST 25438-82 Russian standard (ASTM D1795 and ASTM D4243). The *N*-methylmorpholine-*N*-oxide (NMMO; $H_2O$ < 10%, CAS No. 7529-22-8, Demochem, China) was used as a solvent. Propyl gallate (0.5 wt.%; Sigma-Aldrich, Saint Louis, MO, USA) was added as an antioxidant.

### 2.2. Preparation of Solutions and Films Formation

Films of BC were ground to a powder state with an average particle size of 1–3 mm. Then, the resulting powder was mixed with the solid solvent and antioxidant, and subjected to solid phase activation according to regime described in [26]. The resulting solid phase system was heated to 120 °C and transferred to liquid state. Homogeneity of the solutions was examined by polarization microscopy (Boetius, VEB Kombinat Nadema, former DDR and BIOMED-6PO, Russia).

Films from cellulose solutions were prepared in two stages. In the first stage, the solution was passed through a capillary viscometer Rheoscope 1000 (CEAST, Pianezza, Italy). Further, the layer of the solution was placed between two polyester films coated with hydrophobic siloxane and the sandwich was passed between the rollers of the HLCL-1000 laminator (Cheminstruments, Fairfield, OH, USA) with the fixed gap. To exclude crystallization of NMMO, the temperature of the rollers was maintained in the range 120–125 °C. After formation of the thin layer with desired thickness, the antiadhesive substrates were removed and the solution layer was placed in an aqueous coagulant (T = 22 °C). The final stage of the film preparation was drying to the equilibrium moisture content, which was carried out in isometric conditions.

### 2.3. Characterization Methods

Measurement of the intrinsic viscosity of dilute solutions of BC in cadmium ethylenediamine (cadoxene) was carried out on an Ubbelohde capillary viscometer at 30 °C. The average degree of polymerization was found from the Mark–Kuhn–Houwink equation, with the coefficient $K_m$ and the exponent a, equal to 0.71 and 0.93, respectively (GOST 25438-82 Russian National Standard (ASTM D1795 and ASTM D4243)). As a result, was determined the average value of bacterial cellulose DP,

which is equal to 1500. The experiment was carried out at least three times for each sample and the average intrinsic viscosity value was calculated.

The rheological properties of cellulose solutions were studied on a rotational rheometer MCR 301 (Anton Paar, Austria). A cone-plate operating unit with a diameter of 20 mm (the angle between the cone surface and the plate is 1 degree) was used for measurements. The tests were carried out in steady-state conditions at controlled shear rate in a range of 0.001–1000 s$^{-1}$. Oscillation tests have been done to evaluate the viscoelastic properties of a material in frequency range $10^{-1}$–$10^2$ s$^{-1}$. All measurements were carried out at 120 °C.

The structure of the nascent BC and prepared from solution films was investigated by X-ray diffractometry on Rigaku Rotaflex-RC unit at 30 Kv—100 mA operation mode and CuKα radiation (λ = 1.542 Å). X-ray diffraction measurements were performed in the modes of "transmission" and "reflection" at room temperature. Interplanar distances (d) were calculated using the Wolf-Bragg equation.

The morphology of the nascent BC and spun films were studied by the low-voltage scanning electron microscopy (SEM) using the FEI Scios microscope (USA) with accelerating voltage of less than 1 kV and secondary electrons and Pro X Phenom (Thermo Fisher Scientific, Waltham, MA, USA).

Using the combined thermal analysis instrument TGA/DSC1 (Mettler Toledo, Switzerland), the DSC (Differential scanning calorimetry) and TG curves were obtained in the temperature range of 30–1000 °C at a heating rate of 10 K/min. The flow rate of the argon was 10 cm$^3$/min. This procedure was repeated two times with reproducibility more that 95%.

## 3. Results and Discussion

Photographs and microphotographs of the BC films extracted from the nutrient solution and subjected to treatment with an aqueous solution of NaOH followed by neutralization of the alkali, as well as a comparative morphology of these films are shown in Figure 1.

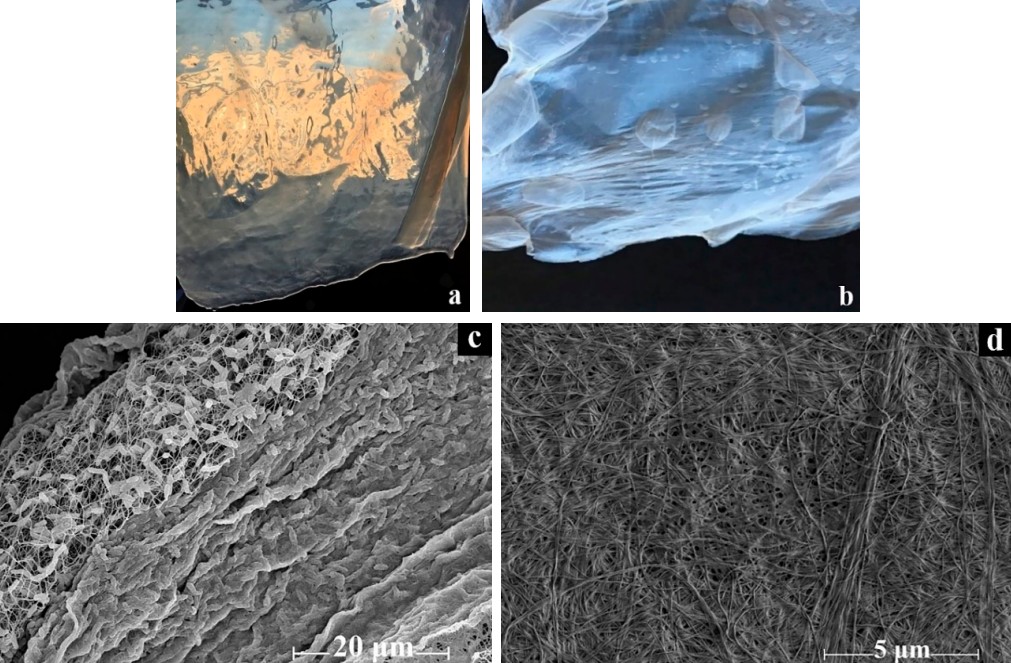

**Figure 1.** Photographs of the nascent bacterial cellulose (BC) film extracted from the mother liquor (**a**), the film after treatment with an alkaline solution and drying (**b**) and microphotographs of the film at different magnifications (**c**,**d**).

It is seen that the BC films remain translucent, both in the wet and dry states, but in the wet state the BC film is more translucent compared with the dry one (Figure 1a,b). Treatment with alkaline solution allows for cleaning the surface of the film from bacteria–producer residues (Figure 1c). As for morphology, the chaotic network of strips is inherent for BC (Figure 1d). The length of the cellulose strips is tens of microns, the average distance between the strips can reach hundreds of nanometers.

X-ray diffractograms obtained in the reflection and transmission modes for wet and dry films of BC are shown in Figure 2.

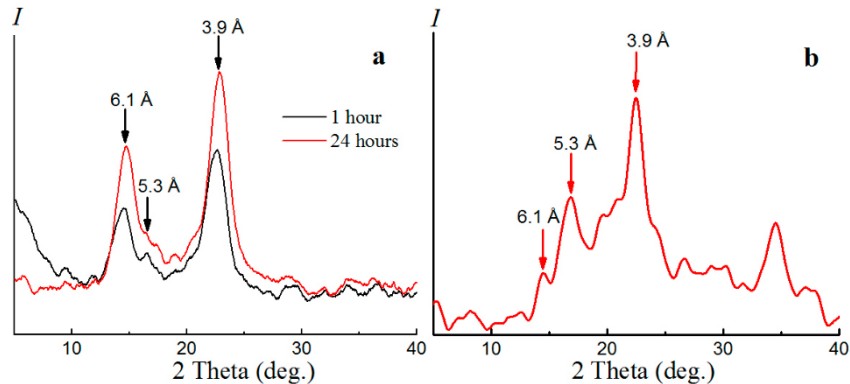

**Figure 2.** X-ray diffractograms of the native BC film in reflection (**a**) and transmission (**b**) modes. Dehydration time: black line—1 h and red line—24 h.

The nascent films were dehydrated under room conditions (T = 22 °C, air humidity ~50%) until the moisture content became equal to ~5%. The evolution of the BC films structure in the process of water removal leads to increase in the intensity of the main cellulose reflexes. The change in the intensity of the basal reflexes during dehydration is due to the change in the preferred orientation of BC crystallites and strips. The absence of a broad background of the water in the $2\theta$ region 20–35° allows for the suggestion that the films lost most of the water during the first hour of dehydration. The angular positions of the basal reflexes (~14.6°, ~16.6° and ~22.7°) correspond to the cellulose polymorphic form I [33]. It is important to note that the intensity of the peaks depends on the diffraction analysis mode. Comparison of the diffraction patterns obtained in reflection and transmission modes revealed a difference in their shape.

Coming to the dissolution of the BC it is useful keep in mind that in the case of cellulose and many other polymers the process proceeds in two stages, swelling and dissolution [21]. Using the optical microscopy it is possible to follow the dissolution process at heating the pre-activated BC-NMMO system to 120 °C and keeping it at this temperature for a definite time until complete dissolution of the polymer (Figure 3). There exists one crucial effect—absence of significant swelling of the sample, but more likely the monotonous decrease of dimensions of the piece.

In the upper left corner inset of the photographs obtained in crossed polaroids are shown in Figure 3a,d, dark circles in the photographs are air bubbles. A series of micrographs (Figure 3a–e) was obtained with a time interval of 1 min.

High activity of the NMMO provides high rates of dissolution of the BC, which allows us to obtain cellulose solutions at short time intervals (Figure 4).

With increasing of BC concentration or its degree of polymerization, the dissolution time increases. In order to reduce the time of solution preparation, it is often necessary to increase the temperature, and this can lead to destructive processes, as can be seen from the data (Figure 4) for $ZnCl_2 \cdot 3H_2O$, where the degree of polymerization catastrophically drops from 4135 to 3200 at 30 °C and to 855 at 95 °C. Authors of [34] have shown that in cellulose solutions in LiCl/DMAc the decomposition of cellulose proceeds at temperature above 85 °C as well. The solution of BC in this solvent with concentration of

8% can be prepared within 40 min, while for NMMO monohydrate the dissolution time increases to 12 h [20].

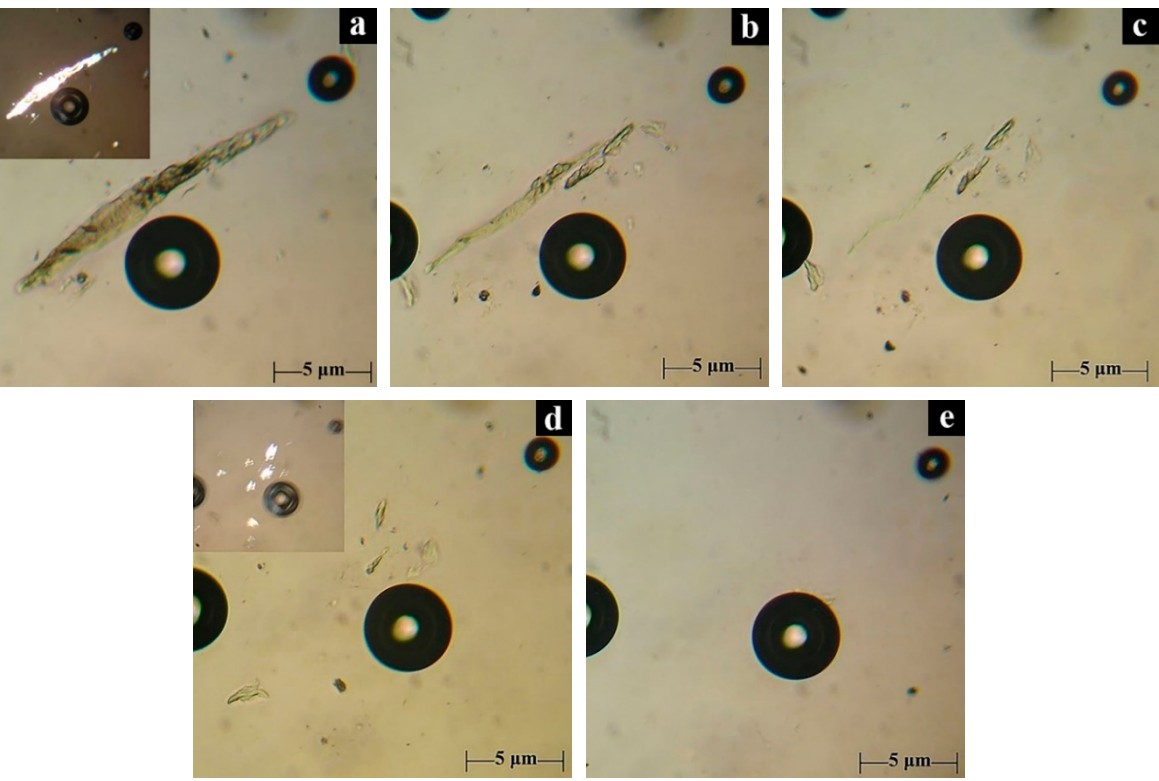

**Figure 3.** Microphotographs of the dissolution of the piece of BC in NMMO (**a–e** were obtained with a time interval of 1 min).

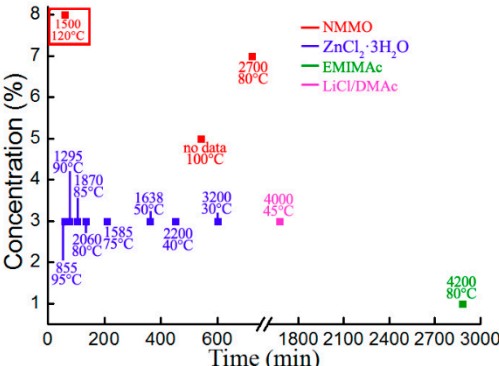

**Figure 4.** Dependence of the BC content passed into solution on the dissolution time for various solvents (different color), temperature and degree of polymerization (DP) of BC (figures close to the experimental points). Data for ionic liquids [17,18], LiCl/DMAc [19], *N*-methylmorpholine-*N*-oxide (NMMO) [10,20,21] and aqueous solutions of zinc chloride [22] as BC solvents are presented.

Although the preparation time of the BC solution in high-melting NMMO after solid state treatment is not crucial since complete dissolution proceeds during 300–2400 s, the change in viscosity (and consequently, DP) over time was estimated (Figure 5). Hydrophilicity of *N*-methylmorpholine-*N*-oxide demands additional requirements for the protection of solutions against moisture in an ambient medium, since water traces play an important role in the phase state of solution and consequently in rheological behavior. To exclude these processes the edge surface of the solution loaded into operating unit contacting with environmental air was coated with a silicone oil ($\eta$ = 0.04 Pa·s at T = 120 °C).

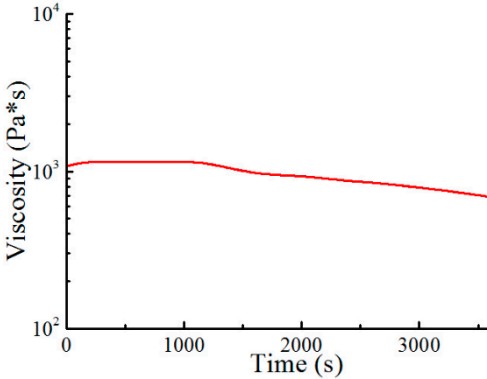

**Figure 5.** Dependence of viscosity ($\gamma = 1\text{s}^{-1}$) on residence time at 120 °C for 6% solution of BC in high melting NMMO.

According to this dependence the viscosity and, accordingly, the cellulose degree of polymerization does not change significantly. The flow curve fragments for solutions of BC with different concentration are presented in Figure 6.

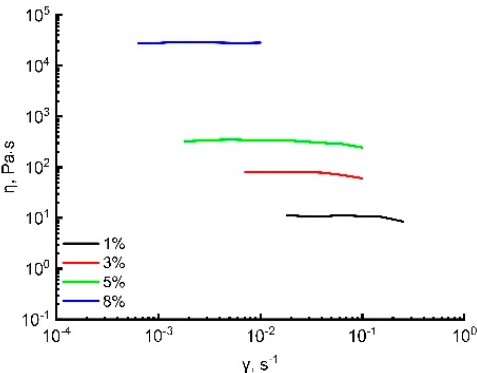

**Figure 6.** Flow curves of solutions of BC in NMMO at 120 °C. Numbers mean the solution concentrations.

In the investigated range of shear rates, solutions of bacterial cellulose in NMMO have almost Newtonian behavior. These results coincide completely with those described in [17], where the rheological properties of the BC solutions in ionic liquids was studied. This kind of behavior was observed in a relatively narrow range of shear rates. With an increase of BC concentration the viscosity increased significantly, which was clearly seen in Figure 7. The figure shows the typical for polymer solutions two lines, intersected at C ~ 4%, with different exponents: 1.54 for low concentration region and ~5 for higher concentrations [17].

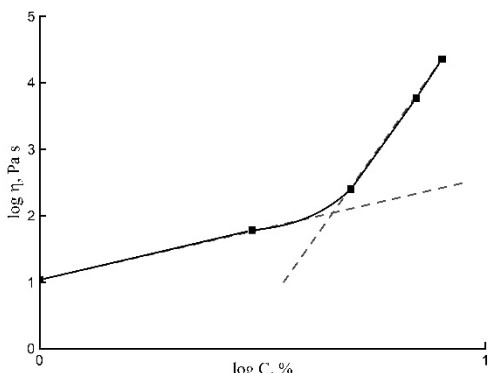

**Figure 7.** Concentration dependence of viscosity for solutions of BC at 120 °C ($\eta = 0.1\ \text{s}^{-1}$).

Data of oscillation experiments are presented in Figure 8.

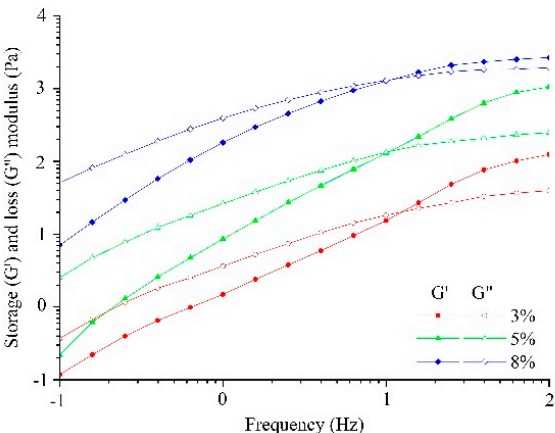

**Figure 8.** Dependences of the storage (filled symbols) and loss moduli (open symbols) on the frequency for BC solutions in NMMO of different concentration.

According to these data, solutions of BC at low concentrations are typical viscoelastic liquids (dissipative modulus is higher than storage modulus). However, at the definite value of frequency the situation changed vice versa and the elastic response prevailed. Tangents of the initial branches of dependences were around 1.4 for G′ and ~0.9 for G″. Increasing the solution concentration from 3% to 8% led to an increase of both moduli on ~two orders.

This kind of behavior was similar to those for the wood cellulose solutions in NMMO.

Based on the results of rheological tests, the spinning solutions (dopes) of bacterial cellulose in NMMO with a concentration around 8% were prepared. The films were formed from the solutions by passing through the calibrated gap between rolls heated to 120 °C. Subsequently, the antiadhesive substrates were removed, and the layer of the BC solution was placed in an aqueous coagulant.

X-ray diffractograms characterizing the structure in the film formed from 8% cellulose solution (after removing of the solvent by water and drying) are presented in Figure 9.

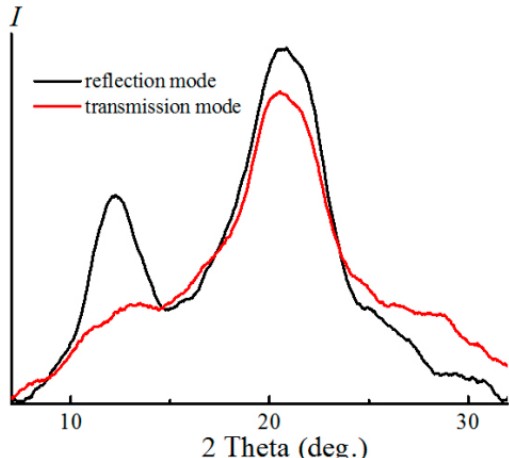

**Figure 9.** X-ray diffractograms of the film formed from 8% solution of BC in NMMO (reflection and transmission modes).

The structure of the film differed from the nascent BC by the angular position of the main peaks, which were located at 2θ ~ 12.1° and ~20.5° compared with 2θ ~ 14.6°, ~16.6° and ~22.7° in the nascent BC. The new positions of the peaks correspond to the cellulose polymorphic form II [33]. It is

reasonable to note that the structure of the nascent cellulose (polymorphic form I) is organized more perfectly, i.e., "bacteria work better than human".

The morphology of the films demonstrated the smooth surface, free from macrodefects (Figure 10b). The rare air microbubbles could only be detected. At large magnifications (100,000×; Figure 10c), some roughness was observed presumably due to micropores formation. The average size of visible pores was 40–50 nm. The most interesting morphology was observed for the cross-section of the films (Figure 10a). The thickness of the obtained films was 2–3 microns. Comparison of morphology of the nascent (Figure 1) and processed films led to the conclusion that during processing the initial fibrils convert into monolithic morphology with longitudinal inclusions formed by residual air with a length of up to 1 micron.

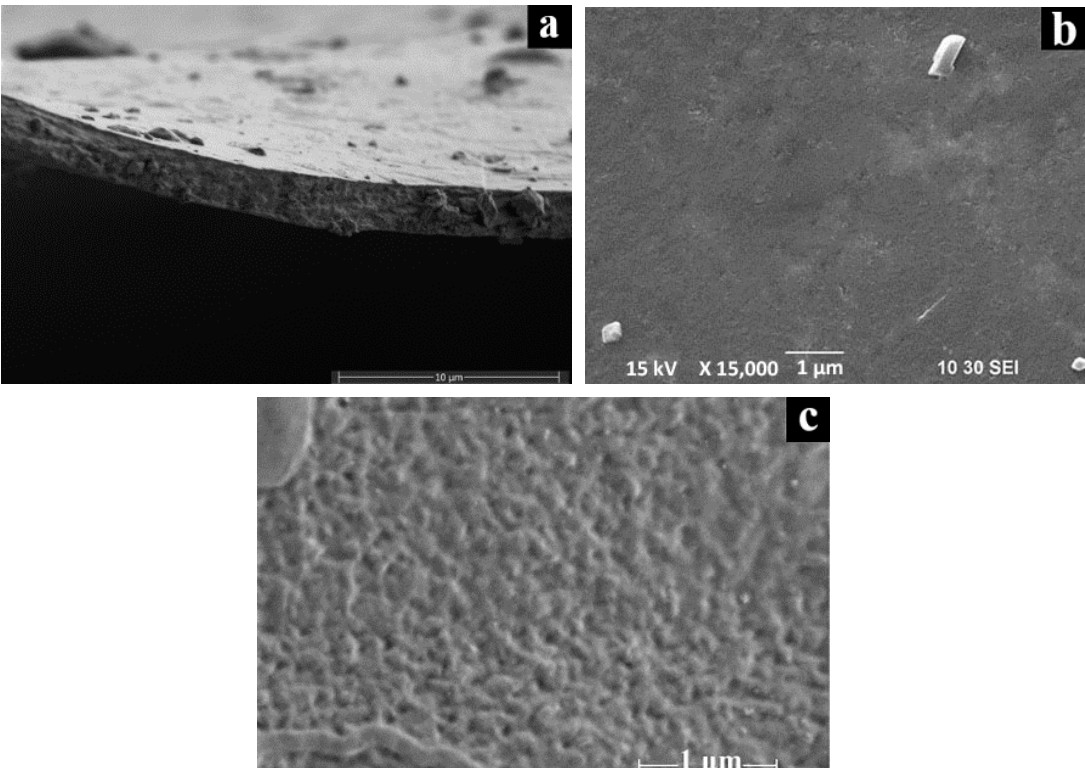

**Figure 10.** Micrographs of regenerated cellulose films (**a**—transverse cross section and **b,c**—surface).

Structural changes occurring in cellulose in the process of processing should undoubtedly affect their properties. To determine water retention and information of thermal decomposition behavior of BC and regenerated BC films thermogravimetric analysis was performed. The neat TG and differential TG (DTG) data are shown in Figure 11.

Three temperature regions could be distinguished on the TG curves. The first region up to 180 °C corresponded to 5% of mass loss. This is due to the release of the adsorbed moisture. An increase of temperature to 350 °C led to a sharp mass loss with a residue of 35% for the initial BC and 43% for the formed film. This region corresponds to processes of dehydration and depolymerization of the polymer [35]. In this case, the pyrolysis of the initial BC started 25 °C earlier than for regenerated film. The rate of mass loss determined by DTG data was higher for the processed films. In the third section of the TG curve (until 1000 °C), where a monotonous loss of mass was observed, the carbon yield at this temperature was 17% for the initial BC and 24% for the formed films. The difference in the values of the carbon yield of the samples being studied was most likely affected by the different structural organization of the films.

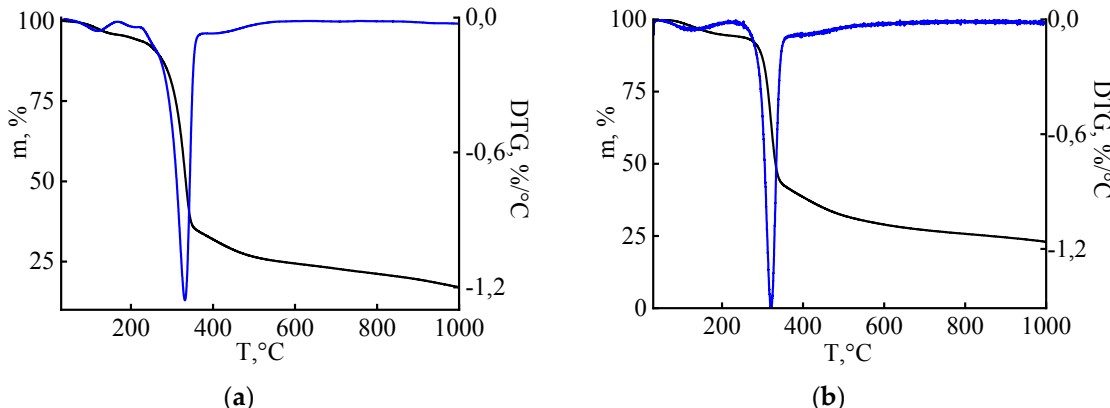

**Figure 11.** TGA and differential TG (DTG) curves of the nascent (**a**) and regenerated (**b**) cellulose films.

It was interesting to see, what kind of the thermal transformations took place for the BC at thermolysis because it did not contain any admixtures, such as lignin, hemicellulose and others. The DSC data can give answer to this question. The corresponding thermograms are shown in Figure 12.

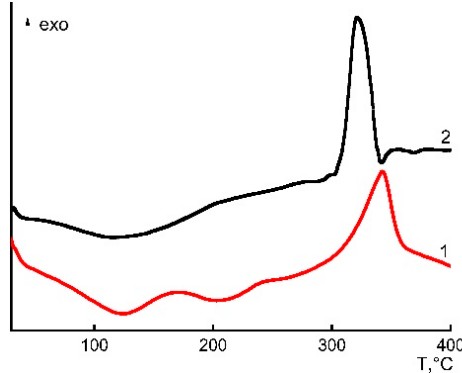

**Figure 12.** Thermograms of the nascent BC (1) and processed (2) films.

In the temperature range corresponding to the removing absorbed water, accompanied by an initial mass loss, the smooth endo-effect was observed. The thermolysis of both films was fundamentally different from the results obtained for the wood cellulose—it is the exo-peak in the range of 290–360 °C instead of the endo-peak [36]. Traditionally, an endo-effect in this temperature range for wood cellulose is considered as destructive processes, but an exo-effect in the case of BC should be considered as ordering processes. It is very difficult now to suggest the mechanism of such kind of ordering. Presumably, all admixtures containing in the wood cellulose can catalyze destruction on various levels. An absence of hemicellulose and lignin in BC renders it possible to change drastically the structure and composition transformation stages. We intended to study these processes in details in the future. It is reasonable to stress that for the nascent BC the exo-effect is shifted to higher temperature compared with processed film. It was not excluded that during processing the structure of the native BC becomes less ordered. Nevertheless, although the order in the system decreased, the latent heat remained the same.

## 4. Conclusions

Cellulose films were formed from concentrated solutions of BC in NMMO via the stage of solid-phase mechano-chemical activation of the powdered components. Comparative studies revealed the advantages of the described method compared to those represented previously in the literature. Thus, solid-phase activation of cellulose with NMMO was responsible for increasing of the dissolution rate of cellulose and as a result of a reduction of the dope time preparation. Short preparation times

of solutions excluded potential destruction processes in the system, preserving the quality of the films and possibility to use again the regenerative solvent. An increase in BC content in the system up to 8% led to a sharp increase in viscosity values ($>10^4$ Pa·s), which imposes restrictions on the processing methods of such solutions. The formed films were characterized by monolithic morphology with a micropore average size of 50 nm, which allows considering these films as potential membrane materials in the future. The thermal behavior of processed BC films was unusual compared with the wood cellulose. The hypothesis about the catalytic role of admixtures at the destruction of the wood cellulose was proposed. It is important to note that completely removing the solvent from the formed films allowed them to be used in the future for medical applications.

**Author Contributions:** I.S.M. and T.I.G. provided the idea for this study, proposed the experiments and wrote the paper; G.K.S. and S.V.L. analyzed the data; V.G.K., I.S.M. and T.I.G. analyzed the data and reviewed the paper; M.I.V. and P.V.Z. produced the samples; G.K.S. prepared the solutions; M.I.V. performed the determination of rheological and thermal properties; N.A.A. investigated the structure of the films; I.S.M. and V.G.K. edited the final paper. All authors have read and agreed to the published version of the manuscript.

**Funding:** This research was funded by Russian Science Foundation, grant number 17-79-30108.

**Conflicts of Interest:** The authors declare no conflict of interest.

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
