# Peer review of "Films of Bacterial Cellulose Prepared from Solutions in N-Methylmorpholine-N-Oxide: Structure and Properties"

_processes, doi:10.3390/pr8020171_

Round 1

Reviewer 1 Report

Dear authors,

Below please find my comments, questions and suggestions to proposed article

I suggest to edit the graphical abstract. First circleplease add what kind of microscope this picture come from, please use italic when you writing latin names of microorganisms, second please provide a name of used strain; Second circle is unclear – what is this for? It shows nothing interesting; Third circle – unclear, it presents nothing interesting, graphical abstract is unacceptable for me. L 17, I suggest to avoid such a strong terms such a „for the first time“ and so on. Please indicate in the abstract the main goal of undertaken experiments. L 29, I suggest to rethink choose keywords. In my opinion they are too general (except bacterial cellulose) so they can not play their role. Why authors did not use for example "N-methylmorpholine-N-oxide"? L 34, I suggest to use abbreviation in this place and after this, just use the abbreviation more often than full term: „...bacterial cellulose (BC) started...“ It fits here better than in line 36. L 35, biocompatibility with what kind of material? Please specify. It is obviously for specialists but your article will be read not only by specialist so please add this information. L 37, I am not sure what authors wanted to say writing: „As a result, a content of cellulose in BC reaches 97%“ Would you be so kind and explain? L 41, BC can be produce not only from fructose so I suggest to increase the range of sugars or leave the term "from some sugars such as: .... and give examples. I also think that using the term „bacterium body“ seems to be clumsy. They do not have bodies. It should be rather „tiny pores of the bacteria cells“ or „tiny pores in bacterial walls“. L42-44, I would rather suggest to remove the sentence: „By analogy (...) coagulation bath“. It does not fit to this text. L 56, please explain the used abbreviation directly after giving full name, and then use abbreviation (I mean DP) L 59, please add a range of this days. L 72-77, I suggest to not use future tense if you present the results which you obtained. In my opinion given one last sentence „This is the aim of this study“ is not proper (again, using present tense is not good in this case).

GENERAL COMMENT to Introduction section: Information about acetic acid bacteria produce BC are highly required as well as examples of using this biopolimer in industry and medicine to highlight the importance of its production.

Again, please strongly indicate the main aim of your study. Please explain why dissolving of BC can be so important, why you provide this procedures and tests?

L 80, please use italic every time you use latin names of microorganisms L 80, please add the source/origin of used strain, patent number or some other information where it comes from. I found this information in ref. 25, but it would be better to put this citation after very first sentence. Please add the information if glucose was the only ingredient in water to create broth? I suppose not, please identify other compounds. Please add the information about pH, time and concentration / volume of inoculum added. Please describe the process of preparing inoculum in details. L 83, 8,33 g BC/1 L –it was a weight of wet or dry cellulose? Please specify. L 84, it should be „the temperature of incubation“ rather that „temperature of the medium“. Please explain: 8,33 g BC on 1 L, and then authors put 13,5 g dry BC on 1 L. What is all about? L 86-87, why it lasted up to 3 days? It is extremelly long. You can find in literature easier and much more short ideas for wash residual cells from BC. Can you provide the references from which you used this method? And please specify what kind of endotoxins you removed from BC? L 88, please specify the concentration of NaOH, please add references for those methods; replace „neutral reaction“ by „neutral pH“ or „pH 7.0“ L 96, detailed description of this grounding is highly required, equipment, parameters, methods of particle measuring, etc. L 97, please explain what exactly means „desired ratio“ L 145, „It is seen, the bacterial cellulose films remain translucent, both in wet and dry states“, yes, it is true, but in wet state BC is more translucent in compare with dry one and it should be underlined. -Please provide information on how many replicates the authors performed their experiments? This is crucial for the preparation of statistical methods that have been omitted in this work In GENERAL Results were described quite well but I feel lack of discussion with results of other authors. In my opinion it is highly appreciate to add more results from other publication to prepare a good discussion with high scientific/knowledge value. It is also very important to prepare statistical analysis of obtained results/datas. We have to make sure that presented results are not only from single samples. L 300, I would rather be careful with using „for the first time“, it is too controversial L304-305, it is not necesarry to repeat methods, please remove this sentence and put CONCLUSIONS. I completly don’t understand what kind of hypothesis are autors writing about? It is first hypothesis in this article and it is in last sentence of all paper!

CONCLUSIONS are NOT CONCLUSIONS at all! They needs to be written from the beginning and include CONCLUSIONS. There is no straight answer on the goal of this study because the goal was not defined. Conclusions in this case are totally unacceptable for m

Author Response

Dear reviewer,

first of all I want to thank you for checking the whole manuscript very carefully. We appreciate it a lot.

I suggest to edit the graphical abstract. First circleplease add what kind of microscope this picture come from, please use italic when you writing latin names of microorganisms, second please provide a name of used strain; Second circle is unclear – what is this for? It shows nothing interesting; Third circle – unclear, it presents nothing interesting, graphical abstract is unacceptable for me. – Done.

L 17, I suggest to avoid such a strong terms such a „for the first time“ and so on. – Done.

Please indicate in the abstract the main goal of undertaken experiments.

- The objective of this paper was to obtain cellulose films from concentrated solutions of BC in NMMO and study their properties. Abstract contains this information - “The films formed from these solutions by means of dry-jet wet spinning in aqueous coagulant were prepared…”.

L 29, I suggest to rethink choose keywords. In my opinion they are too general (except bacterial cellulose) so they can not play their role. Why authors did not use for example "N-methylmorpholine-N-oxide"? - Done

L 34, I suggest to use abbreviation in this place and after this, just use the abbreviation more often than full term: „...bacterial cellulose (BC) started...“ It fits here better than in line 36. - Thank you very much for your comments. - Done.

L 35, biocompatibility with what kind of material? Please specify. It is obviously for specialists but your article will be read not only by specialist so please add this information. - Done

L 37, I am not sure what authors wanted to say writing: „As a result, a content of cellulose in BC reaches 97%“ Would you be so kind and explain? - There was a sign alpha (alpha-cellulose)

L 41, BC can be produce not only from fructose so I suggest to increase the range of sugars or leave the term "from some sugars such as: .... and give examples. I also think that using the term „bacterium body“ seems to be clumsy. They do not have bodies. It should be rather „tiny pores of the bacteria cells“ or „tiny pores in bacterial walls“. – Thank you very much for precious comments. Done.

L42-44, I would rather suggest to remove the sentence: „By analogy (...) coagulation bath“. It does not fit to this text. – Done.

L 56, please explain the used abbreviation directly after giving full name, and then use abbreviation (I mean DP) – Done.

L 59, please add a range of this days. – Done.

L 72-77, I suggest to not use future tense if you present the results which you obtained. In my opinion given one last sentence „This is the aim of this study“ is not proper (again, using present tense is not good in this case). – Done.

GENERAL COMMENT to Introduction section: Information about acetic acid bacteria produce BC are highly required as well as examples of using this biopolimer in industry and medicine to highlight the importance of its production. – Done.

Again, please strongly indicate the main aim of your study. Please explain why dissolving of BC can be so important, why you provide this procedures and tests? – Done.

L 80, please use italic every time you use latin names of microorganisms – Done.

L 80, please add the source/origin of used strain, patent number or some other information where it comes from. I found this information in ref. 25, but it would be better to put this citation after very first sentence.  – Done.

Please add the information if glucose was the only ingredient in water to create broth? I suppose not, please identify other compounds. Please add the information about pH, time and concentration / volume of inoculum added. Please describe the process of preparing inoculum in details. – Done.

L 83, 8,33 g BC/1 L –it was a weight of wet or dry cellulose? Please specify. – Dry cellulose

L 84, it should be „the temperature of incubation“ rather that „temperature of the medium“.  -Done

Please explain: 8,33 g BC on 1 L, and then authors put 13,5 g dry BC on 1 L. What is all about?

- The strain synthesizes such an amount on a standard medium of M. Schramm and S. Hestrin described in [Schramm M., Gromet Z., Hestrin S. // Biochem J. 1957.V. 67. â„– 4. P. 669–679.]. On a modified nutrient medium, the strain synthesizes 13.0 ± 0.5 g / l.

L 86-87, why it lasted up to 3 days? It is extremelly long. You can find in literature easier and much more short ideas for wash residual cells from BC. Can you provide the references from which you used this method? And please specify what kind of endotoxins you removed from BC? – Done.

State Quality Standards of Medicines, General pharmacopoeia monograph, 42-0002-00,

Bacterial endotoxins, 26 December, 2000 (in Russian).

US (2012) ‘Guidance for industry – pyrogen and endotoxins testing’, Services USDoHH (Ed.),

U.S. Food and Drug Administration, Silver Spring, June, p.11.

L 88, please specify the concentration of NaOH, please add references for those methods; replace „neutral reaction“ by „neutral pH“ or „pH 7.0“ – Done.

L 96, detailed description of this grounding is highly required, equipment, parameters, methods of particle measuring, etc. L 97, please explain what exactly means „desired ratio“  - Done. “in the desired ratio” was removed.

L 145, „It is seen, the bacterial cellulose films remain translucent, both in wet and dry states“, yes, it is true, but in wet state BC is more translucent in compare with dry one and it should be underlined. - Thank you very much for your comment. This information has been added.

-Please provide information on how many replicates the authors performed their experiments? This is crucial for the preparation of statistical methods that have been omitted in this work – Done.

In GENERAL Results were described quite well but I feel lack of discussion with results of other authors. In my opinion it is highly appreciate to add more results from other publication to prepare a good discussion with high scientific/knowledge value. It is also very important to prepare statistical analysis of obtained results/datas. We have to make sure that presented results are not only from single samples.

- Thank you very much for your suggestion. We did a full literature analysis on BC direct solvents, the technological parameters of obtaining solutions of BC, which is reflected in the introduction and the experimental part. As can be seen from the presented fig. 4, none of the generalized solvents allows to obtain solutions with a cellulose concentration of more than 5%. As for NMMO, it was our group that was able to obtain concentrated solutions in short periods of time, thereby avoiding its destruction [34]. Further, all the results obtained for BC solutions, the structures of the resulting films, and their properties are correlated with published data.

L 300, I would rather be careful with using „for the first time“, it is too controversial. - Done.

L304-305, it is not necesarry to repeat methods, please remove this sentence and put CONCLUSIONS. I completly don’t understand what kind of hypothesis are autors writing about? It is first hypothesis in this article and it is in last sentence of all paper! CONCLUSIONS are NOT CONCLUSIONS at all! They needs to be written from the beginning and include CONCLUSIONS. There is no straight answer on the goal of this study because the goal was not defined. Conclusions in this case are totally unacceptable for m – Done.

Best regards,

Igor Makarov

Reviewer 2 Report

Dear Authors,

my comment:

Abstract: Please give more information about the Results in Abstract Section

Line 56: what is a DP?

Line 77: Rewrite this sentence.

Line 181: Figures 4 change Fig.4

Author Response

Dear reviewer,

Thank you very much. We appreciate your recognition.

Abstract: Please give more information about the Results in Abstract Section

- Done

Line 56: what is a DP?

- Thank you very much for your comment. We specified abbreviation DP.

Line 77: Rewrite this sentence.

- Thank you very much for your suggestion. The sentence has been rewrited.

Line 181: Figures 4 change Fig.4

- Done

Best regards,

Dr. Igor Makarov

Round 2

Reviewer 1 Report

Dear Authors,  thank you for your corrects. I accept your paper in this form and wish you further scientific success.